# “The Two Sides of the Same Coin”—Medical Cannabis, Cannabinoids and Immunity: Pros and Cons Explained

**DOI:** 10.3390/pharmaceutics14020389

**Published:** 2022-02-10

**Authors:** Mona Khoury, Idan Cohen, Gil Bar-Sela

**Affiliations:** 1Cancer Center, Emek Medical Center, 21 Yitzhak Rabin Blvd, Afula 1834111, Israel; mona_kh1994@hotmail.com (M.K.); idan5161@gmail.com (I.C.); 2Bruce Rappaport Faculty of Medicine, Technion-Israel Institute of Technology, Haifa 3200002, Israel; 3Oncology & Hematology Division, Emek Medical Center, Yitshak Rabin Boulevard 21, Afula 1834111, Israel

**Keywords:** medical cannabis, endocannabinoids, phytocannabinoids

## Abstract

Cannabis, as a natural medicinal remedy, has long been used for palliative treatment to alleviate the side effects caused by diseases. Cannabis-based products isolated from plant extracts exhibit potent immunoregulatory properties, reducing chronic inflammatory processes and providing much needed pain relief. They are a proven effective solution for treatment-based side effects, easing the resulting symptoms of the disease. However, we discuss the fact that cannabis use may promote the progression of a range of malignancies, interfere with anti-cancer immunotherapy, or increase susceptibility to viral infections and transmission. Most cannabis preparations or isolated active components cause an overall potent immunosuppressive impact among users, posing a considerable hazard to patients with suppressed or compromised immune systems. In this review, current knowledge and perceptions of cannabis or cannabinoids and their impact on various immune-system components will be discussed as the “two sides of the same coin” or “double-edged sword”, referring to something that can have both favorable and unfavorable consequences. We propose that much is still unknown about adverse reactions to its use, and its integration with medical treatment should be conducted cautiously with consideration of the individual patient, effector cells, microenvironment, and the immune system.

## 1. Introduction

The relationship between man and the Cannabis plant reaches back thousands of years; its first known use is as early as 10,000 years ago and as far back as the Stone Age [1]. The first historical evidence for the spiritual and medical uses of cannabis is from central Asia, India, and China, where its primary use was in treating rheumatic pain, malaria, beriberi, and gynecological disorders [2]. Cannabis use spread across Asia to Japan, Iran, Egypt, and the Eastern medical world [3]. It continued forward, reaching the Western world around AD 1000–1200 when cannabis seeds were found in the remains of Viking ships dating to the mid-ninth century [4]. Although the medical use of cannabis was applied in many diseases freely [2], the plant was outlawed in Utah in 1915 and, by 1931, it was illegal in 29 states across the U.S. In 1937, the Marijuana Tax Act finally placed cannabis under the regulation of the Drug Enforcement Agency, criminalizing possession of the plant throughout the country, and was removed from the U.S. National Formulary and Pharmacopeia [3]. While considered a dangerous drug in many countries, several pioneers continued to research the plant throughout the 20th century, unveiling its multiple plant-based cannabinoids, endogenous cannabinoid receptors, and several other endocannabinoid system elements [5]. In recent years, many countries have legalized medical cannabis to treat various diseases [6]. Today, the term “medical cannabis” is a popular label for a group of pharmacological products derived from the cannabis plant, referring to the use of dried leaves, flowers, stems, and seeds, and including several types of plant extracts.

The term ‘cannabinoids’ actually describes a family of compounds divided into three groups: endocannabinoids are naturally formed signaling endogenous ligands that regulate various physiological conditions via cannabinoid receptors; phytocannabinoids are naturally occurring active compounds extracted from the *Cannabis sativa* plant able to bind and mimick endocannabinoid signaling by interacting with the same cannabinoid receptors; and synthetic cannabinoids (such as dronabinol and nabilone) are chemically synthesized artificial molecules designed for medicinal use to interact with, activate or inhibit the same cannabinoid receptors that mimic the effects of cannabinoids. Nowadays, 545 active natural compounds have been extracted and characterized from cannabis, including phytocannabinoids, terpenes, hydrocarbons, sugar, nitrogenous compounds, non-cannabinoid phenols, fatty acids, simple acids, and flavonoids [7]. Currently, at least 144 natural phytocannabinoids are known, with the most abundant being tetrahydrocannabinol (THC) and cannabidiol (CBD) [8].

The five main methods of cannabis usage are: orally (slowly dissolving tabs placed under the tongue or swallowed in pill form), non-combustion inhalation (E-cigarettes, vapor pens, vapor machines), combustion inhalation (tobacco-based cigarettes commonly referred to as joints and bungs), in an ointment form, and also can be combusted and inhaled alone (without tobacco). Each of these techniques has its unique qualities, no data favors any one method, and treatment choice is individual. However, cannabis combustion is considered more problematic and less favorable than vaporization due to exposure to the harmful toxic substances created by combustion. Cancer, autoimmune, inflammatory bowel diseases, and autism, to name but a few, are conditions and symptoms for which cannabis is used as a palliative treatment [9,10,11,12].

Cannabis can suppress the body’s immune system, yet this is still mainly overlooked and ignored [13,14]. Research and clinical discoveries uncovering the effects of cannabis on the immune system emphasize the urgent need to investigate further the impact of cannabis use on immune cells as an accompanying popular treatment, with a need to communicate the data, findings, and information to physicians, consumers, and patients.

## 2. Medical Uses of *Cannabis sativa*

Today, as a palliative treatment, we know that cannabis and its sedative properties are very effective in combating drug and treatment side effects for various medical conditions. It has been approved or is under consideration for legalization in many countries worldwide and is understood to be safe relative to opioids [15,16,17,18].

### 2.1. Cancer-Related Symptoms

A recent uncontrolled prospective study of a large group of cancer patients showed a significant improvement of the common symptoms when introduced to cannabis, such as insomnia (70.8%), anxiety and depression (74.1%), nausea and vomiting (54.7%) [19,20], demonstrating the proven clinical efficiency of cannabis as a palliative treatment in anti-cancer therapy [19,20,21,22].

### 2.2. Neuropathic, Neurological and HIV-Related Symptoms

Worldwide, chronic pain is a general therapeutic challenge, affecting more than 15% of the population [23]. While opioid-based drugs manage severe pain well, their adverse effects, mortality rates, and the tendency for addiction are primary concerns. As evident by clinical trials, cannabis’ therapeutic benefits in the neuropathy of pain when using nabiximols (sativex), for example, a specific cannabis extract approved in 2010 as a botanical drug in the United Kingdom, containing THC and CBD in an equal ratio, aided in pain management [24,25,26,27]. In a sharp difference to opioid-based drugs, nabiximols were shown to be well tolerated in clinical trials with very mild side effects, such as dizziness or disorientation with no significant adverse effects on cognition [28,29,30]. Although mostly based on non-randomized studies with no comparators, a recent systematic review also shows that using cannabis by patients with post-traumatic stress disorder (PTSD) improves pain and sleep and is associated with reduced overall PTSD symptoms and reports of a better quality of life [31,32]. Likewise, cannabis provides a beneficial intervention for reducing inflammation-related morbidity in acquired immunodeficiency syndrome (AIDS) patients [33], also improving muscle and nerve pain, and may also assist with depression [34,35,36]. In multiple sclerosis patients, cannabis reduces spasticity compared to placebo [37,38,39]. Cannabis and cannabinoids are also effective in yielding a significant positive effect on epileptic seizure load, improving behavior and alertness, language, communication, motor skills, and patients’ sleep or allowing muscle relaxation and pain relief in Parkinson’s disease patients [40,41,42]. A significant improvement was also perceived in insomnia by cannabinoid use [43].

Taken together, looking at the safety and efficacy of cannabis consumption and cannabis-based products in clinical trials, there is no doubt about the benefits of cannabis-based therapy in the aid and ease of disease symptoms and treatment side effects. The cumulative data throughout the years also support the therapeutic potential of cannabis integration and utilization in other diseases. Patients dealing with various conditions are able to enjoy the positive “one side of the coin” of cannabis and cannabinoids and be assisted by its wide range of treatment qualities.

## 3. The Endocannabinoid System (ECS)

The endocannabinoid system (ECS) is an essential signaling system in mammal physiology regulating several biological and disease conditions. ECS’s three main components are (i) cannabinoids receptors (CBR) CBR-1 and CBR-2; (ii) signaling molecules that are lipid-based termed the “endocannabinoids” (EC), endogenous ligands of the CBR’s; and (iii) enzymes responsible for synthesizing and degrading ECs. Over decades, intensive research on the subject has seemingly brought to light that the ECS regulates multiple physiological processes, such as brain plasticity and neuronal development, cell death, inflammation, sleep, appetite, pain, and anxiety [44].

ECs are comprised of highly lipophilic ligands derived from the fatty acid-arachidonic acid (AA), and the first identified EC is called anandamide, also known as N-arachidonoylethanolamine (AEA) and 2-arachidonoyl glycerol (2-AG) [45,46]. Initially isolated from the porcine brain, anandamide is a partial agonist with CBR-1 and CBR-2. Next, 2-AG was isolated from canine intestines as a full agonist with CBR-1 and CBR-2 [47,48,49]. Further putative ECs have been discovered more recently. For example, the EC 2-arachidonoyl glycerol ether (2-AGE, Noladin ether) binds to CBR-1, and CBR-2, showing agonistic properties to both receptors and partial agonistic characters to the TRPV1 channel (Figure 1) [50,51]. The first discovered antagonist for CBR (CBR-1) was Virodhamine (O-AEA), even though its physiological role has yet to be fully understood (Figure 1). [52]. At a later stage, the enzymes, fatty acid amide hydrolase (FAAH) and monoacylglycerol lipase (MAGLs), responsible for the degradation of ECs, were further discovered and characterized [53].

### 3.1. ECS and CBRs—Regulating Immune Functions

The effects of the ECs are primarily mediated by CBR-1 and CBR-2 receptors when the impact of other recently discovered cannabinoid receptors, such as peroxisome proliferator-activated receptors (PPARs) and transient receptor potential channels (TRP channels), are not yet fully interpreted [54]. CBR-1 and CBR-2 are G-protein-coupled receptors; their activation inhibits adenylyl cyclase and specific voltage-dependent calcium channels and activates mitogen-activated protein kinase (MAPK) [54]. Therefore, the activation of CBR’s has diverse consequences on cellular physiology, synaptic function gene transcription [54].

In 1988, using a radiolabelled ligand specifically binding to the CBR, the CBR-1 receptor was discovered [55]. CBR-1 is primarily expressed in the mammalian brain, including the cortex, basal ganglia, hippocampus, and cerebellum [56]. The widespread distribution of the CBR-1 is associated with regulating various cognitive and neuronal functions, ranging from memory, mood, appetite, and sensory responses [57]. Nonetheless, CBR-1 is also expressed at low levels in peripheral tissues and cells [58], and its expression can be found in the adrenal gland, heart, liver, lung, and other peripheral tissues [59,60]. The activation of the presynaptic CBR-1 inhibits the release of neurotransmitters from the synapses by decreasing calcium conductance and increasing potassium conductance [61].

In contrast to CBR-1, the CBR-2 receptor is expressed at a low level in the central nervous system (CNS). CBR-2 is primarily expressed in the spleen, tonsils, thymus, and other tissues and is believed to be responsible mainly for regulating the immune system [58,62]. CBR-2 mRNA expression differs between immune system cells and is expressed at the following hierarchy: B cells > natural killer (NK) cells > polymorphonuclear (PMN) > neutrophils > CD8+ T cells > monocytes > CD4+ T cells [58]. The observation that peripheral blood mononuclear cells (PBMC) from marijuana smokers elevate the expression of CBR receptors compared to healthy donors [63] suggested that CBR’s expression levels on immune cells may be sensitive and adjusted by exposure to phytocannabinoids. On the same line, cytokines could also regulate the CBRs expression [64], while transforming growth factor-beta (TGF-β) regulated CBR-2 expression in human peripheral blood lymphocytes (PBL) via a negative autocrine regulatory loop and IFN-γ produced by T helper 1 (Th1) and NK cells increase the expression of CBR-2 in rat macrophages in neuropathic pain or multiple sclerosis (MS) animal models [65,66]. Accordingly, CBR-1-deficient mice cannot develop lipopolysaccharide-evoked fever, have a lower expression of Toll-like receptor 4 (TLR4) than wild-type (WT) mice, and their peritoneal macrophages do not secrete pro-inflammatory cytokines in response to lipopolysaccharide (LPS) [67]. Conversely, CBR-2 expression protects against acute experimental sepsis when CB2-R deletion (CB2R(^−/−^)) sensitizes mice to LPS induced death, and the CBR-2 agonist, GW405833, could significantly extend their survival [68].

A few other receptors have recently been identified and characterized as cannabinoids interacting and binding receptors. PPARs are a family of nuclear receptors that include PPARα, PPARγ, and PPARβ/δ, acting as transcription factors primarily involved in the transcriptional regulation of genes linked with inflammation and metabolism [69]. The increased transcriptional activity of PPARα after binding the endocannabinoids 2-AG and oleoyl ethanolamide (OEA) has been shown to regulate genes associated with metabolism and anti-inflammatory effects (Figure 1) [70]. Similarly, different phytocannabinoids can bind, interact, and regulate PPAR activity when natural and synthetic phytocannabinoids, including THC, CBD, cannabimovone (CBM), and quinone derivatives, are all characterized as PPARγ agonists (Figure 1) [70,71].

TRP channels, known as modulators of ion entry, mediate various neuronal signals controlling the sensation of pressure, temperature, and smell and are also regulated by cannabinoids [72]. In recent years, it has come to light that the TRP channel also retains a functional extra-neuronal expression in immune and epithelial cells with important implications for mucosal immunology. For example, TRP channels maintain intracellular calcium homeostasis, regulating various immunological functions, such as producing and releasing inflammatory mediators, phagocytosis, or even immune-cell migration [73]. Correspondingly with the role of ECS in immunity, AEA was identified as an endogenous agonist of TRPV1 [74], while later, N-arachidonoyl dopamine (NADA) and AEA were also characterized as TRPM8 antagonists (Figure 1) [75]. Furthermore, phytocannabinoids can bind and regulate TRPV channels when CBD is a potent agonist at TRPV1 and TRPM8 (Figure 1) [76], while delta9-THC mainly interacts with TRPV2 but also moderately modulates with TRPV3, TRPV4, TRPA1, and TRPM8 activities (Figure 1) [72,76]. Thus, understanding EC’s mechanism in influencing these channels is essential for various immunological and therapeutic processes, such as inflammation and its resolution and host immunity [72].

The G protein-coupled receptors (GPCRs) are prevalent and popular pharmaceutical targets for drug design, representing one of the largest and most versatile cell surface receptors family [77]. The activation of GPCRs is achieved by a diverse range of ligands, including chemokines, lipids, vitamins, hormones, protein, peptides, and neurotransmitters [78]. Therefore, GPCRs are involved in many fundamental cellular and physiological processes, such as the ability to sense light, taste and smell, metabolism, endocrine, and exocrine secretion [79].

In recent years, GPCRs have been found to play an important role also in the immune system [79]. GPCRs, such as GPR18, GPR55, and GPR119, regulate signals associated with cell proliferation and migration [80]. Moreover, GPR18 is expressed on human leukocytes and is believed to be responsible for regulating immune system cells during inflammation [81]. The endocannabinoid N-arachidonoyl-l-serine (ARA-S) mediates angiogenesis and wound healing through GPR-55 (Figure 1) [82]. CBD in a mouse model of dravet syndrome shows antagonism of GPR-55 effectively reduced seizures [83].

### 3.2. Phytocannabinoids and Their Use in Medicine

Phytocannabinoids are bioactive natural products found in some flowering plants, liverworts, and fungi and can be used to treat diseases or undesirable physiological symptoms [84]. *Cannabis sativa* is an abundant and most studied source of phytocannabinoids [84]. Currently, approximately 130 distinct phytocannabinoids have been isolated from cannabis plants and classified into several groups: cannabigerols (CBGs), cannabichromenes (CBCs), cannabidiols (CBDs), (−)-delta9-trans-tetrahydrocannabinols (delta9-THCs), (−)-delta8-trans-tetrahydrocannabinols (delta8-THCs), cannabicyclols (CBLs), cannabielsoins (CBEs), cannabinols (CBNs), cannabinodiols (CBNDs), cannabitriols (CBTs), and a few other miscellaneous phytocannabinoids [84]. The most abundant *Cannabis sativa* constituents are trans-delta 9-THC, CBD, CBC, and CBG, together with their respective acid forms (delta9-tetrahydrocannabinolic acid A (delta9-THCA), CBDA, CBCA, and CBGA) [84]. However, it is essential to note that phytocannabinoids and ECs mediate their effects as antagonists or agonists by interacting with the same specific cell-surface CBRs, essentially acting as ECs.

To date, the most utilized, popular, and intensively studied phytocannabinoids are delta9-THC and CBD, although lately, other phytocannabinoids derivatives, such as Δ9-tetrahydrocannabivarin (delta9-THCV), cannabinol (CBN), cannabidivarin (CBDV), cannabigerol (CBG), and cannabichromene (CBC), have become of significant interest [85]. Discovered in 1971, THC, the primary psychoactive compound of cannabis, was initially not widely used, but later led to the discovery of CBR-1 and a better definition of the ECS [86]. Today, THC is an approved medical therapy drug [87]. Nabiximols, a cannabis-based preparation containing a standardized ratio of THC and CBD, is an approved treatment for people with MS and has been used in the United Kingdom since 2010, then in Spain, Germany, Denmark, Sweden, Italy, Austria, and Poland since 2011–2012. France legalized cannabinoids in medicine in 2013 [88], and the brand name Sativex has been approved in Canada for cancer-related pain and was recently legalized in Ukraine. Moreover, THC products are commonly used as an effective remedy in a range of neurodegenerative and neurological disorders, such as Huntington, Parkinson’s, Alzheimer’s, and Tourette syndromes [89]. In addition, currently in the U.S., there are two chemically synthesized, cannabis-based drugs approved by the U.S. Food and Drug Administration (FDA) for use primarily in oncological patients. Dronabinol (Marinol) is a gelatin capsule containing an isomer of THC, used to relieve nausea and vomiting caused by chemotherapy or enhance the appetite to achieve weight gain in AIDS patients. Although it is a synthetic-based cannabinoid, Nabilone (Cesamet), acting much as THC, is another FDA-approved alternative drug to treat nausea and vomiting caused by chemotherapy [90].

CBD, the second most abundant phytocannabinoid, accounts for up to 40% of the plant’s extract and shows almost no affinity for CBR-1 and CBR-2, but it can antagonize the effect of THC and AEA to CBR-1 [91]. CBD does not have the same psychoactivity as THC and, thus far, CBD oil is mainly used to treat pain by affecting non-cannabinoid receptors, such as TRP channels and PPARs [91]. Although the U.S. removed hemp and hemp extracts (including CBD) from the list of controlled substances in 2018, and the FDA approved the first CBD-containing medical product, Epidiolex^®^, for the treatment of two rare epilepsy disorders, proving the tremendous medical potential of cannabis-based products [92], the marketing and sales of CBD formulations remain a grey area in many states. Contrary to the U.S., European Union member states allow CBD products legally sold as an antioxidant, anti-sebum, skin protection, and skincare (cosmetics) product, but they are restricted from being marketed as medical products improving health [92].

## 4. Medical Cannabis Phytocannabinoids and the Immune System

In the human immune system, immune cells recognize invading microbial pathogens; the immune response to infection depends on the release of various pro-inflammatory cytokines, metabolites, and proteins [93]. Since many of the cannabinoid and cannabinoid-like or putative orphan receptors are transcribed and expressed as surface receptors on most innate and adaptive immune system members [81,94], it is expected that ECS and phytocannabinoids play a role in regulating and modulating inflammatory and cellular immune processes.

Aiming to understand the role of ECS in regulating the immune system during inflammation, anti-cancer, and infection, numerous in vitro and in vivo studies have shown that EC signaling and production by immune cells can modulate a wide range of immune responses, e.g., increased production of 2-AG by human dendritic cells (DCs) after LPS was reported [95,96]. At the same time, 2-AG also demonstrates chemoattractant properties inducing the migration of human eosinophils, dendritic or B- and NK cells [97,98,99,100].

In mammals, phytocannabinoids also demonstrate a potent anti-inflammatory effect in CBR’s mediated pathways [101]. For example, in an ex vivo exposure of thymocytes and naive and activated splenocytes to 10–20 µM, THC increased apoptosis of immune cell populations, such as T and B cells, macrophages, and DCs (Table 1) [102]. Moreover, the in vivo administration of 10 mg/kg body weight of THC into C57BL/6 mice led to thymic and splenic atrophy with reductions in splenocyte and thymocyte subpopulations counts. Likewise, splenocytes from THC-treated mice significantly reduced proliferation in response to different stimuli, and both thymocytes and splenocytes exhibited apoptosis upon in vitro culture. Although mostly by ex vivo studies with primary mouse cells, THC (15–30 μM) induced apoptosis in macrophages (Table 1) [103].

In other support of this hypothesis, mice administered with THC in vivo (1–50 mg/kg) showed decreased splenic DCs, and MHCII expression by DCs triggered the NF-kappaB-dependent mechanism (Table 1) [101,104]. These findings led the authors to suggest that in vivo and in vitro exposure to THC can lead to a significant immuno-suppression effect by induction of immune cells apoptosis [102]. CBD also induced apoptosis in CD4+ and CD8+ T cells at 4–8-μM concentrations by increasing reactive oxygen species (ROS) production, as well as caspase 3 and 8 activity. Synthetic cannabinoids have also induced apoptosis of human T lymphocytes (Table 1) [101]. It would seem that one of the central processes by which phytocannabinoids exert their potent immunosuppressive effect is by inducing cell death or apoptosis of immune cell populations.

Aside from cell death, suppressing cytokine production also plays a central role in phytocannabinoid immunosuppression properties. The administration of CBD 5 mg/kg reduces in vivo production of male adult Wistar rats serum cytokines (Table 1) [105]; when exposed to THC or CBD in vitro experiments, human eosinophils T, B, CD-8, and NK cells decrease cytokine production after activation (Table 1) [106]. A novel *Cannabis sativa* L. extract standardized in 5% CBD with a low content of THC (<0.2%) recently inhibited human neutrophil functions, such as migration, ROS generation, and TNF-α production (Table 1) [107]. Recently, in a clinical study, we were able to show that cannabis consumption by advanced cancer patients indeed alleviated the disease symptoms and side effects; however, it could also impair the chances of the immunotherapy given to activate the patients’ immune system against the tumors, directly impacting tumor response rates and survival [90,108,109].

A description and review of the direct effects of different phytocannabinoids on the immune system’s effector cells in patients using cannabis will follow our aim to highlight and emphasize the possible detrimental impact of these plant-based products.

### 4.1. T Lymphocyte Regulation by Cannabinoids

T lymphocytes express antigen-specific cell-surface receptors that allow recognition of different pathogens and comprise a significant source of various cytokines. T lymphocytes can also recognize normal tissue during episodes of autoimmune diseases. Two main subsets of T lymphocytes are distinguished by the surface molecule CD4 and CD8 [110]. T lymphocytes expressing CD4 are also known as helper T cells and are sub-divided into Th1 and Th2 [110].

An in vivo study gave the first indication that T lymphocytes were sensitive and regulated by cannabinoids and cannabis consumption, showing a reduction in T-cell proliferation in healthy controls who smoked marijuana [111]. In agreement with this observation, phytocannabinoid compounds, such as THC, CBN, CBD and CBRs, regulate cytokine production and proliferation of T cells (Table 1) [112,113,114]. The notion that CBRs are involved in regulating T-cell functions can be seen by the fact that CD8+ T lymphocytes migration is inhibited via the CBR-2 [115]. Activated splenocytes derived from CB1^−/−^ CB2^−/−^ mice also showed reduced IL-2 and IFN-γ cytokines production [112]. THC, in particular, was seen to suppress the proliferation of spleen and lymph node cells of adult and young mice (Table 1) [116], while in another murine model, THC was seen to downregulate T helper 1-(Th1) associated cytokines, including g IL-2, IL-12, and interferon-g (IFN-g), increasing the production of T helper 2 (Th2)-related cytokines, such as IL-4 and IL-10, and also to suppress the immune response to Legionella pneumophila infections (Table 1) [117,118].

The immunosuppressive effects of THC and other agonists of CBR-1 and CBR-2 are further demonstrated by leading to a massive increase in cAMP, which inhibits the subsequent T-cell receptor cascade and IL-2 production (Table 1) [119]. In parallel, CBD (16μM) treatment, ex vivo and in vitro, in thymocytes derived from BALB/c mice and EL-4 thymoma cell lines, induced apoptosis in CD4 and CD8 T cells by increasing ROS generation (Table 1) [120]. Likewise, CBD suppresses IL-2 and IFN-γ cytokines [112] when CBN enhances IL-2 production in phorbol-12-myristate-13-acetate (PMA) activated T cells, both in a CBR-1/CBR-2-independent manner [121]. CBD also showed anti-proliferative properties, inhibiting myelin oligodendrocyte glycoprotein (MOG) induced T-cell proliferation in vitro when neither a CBR-1 antagonist nor a CBR-2 antagonist affected the anti-proliferative effect of CBD [122].

Collectively, the compiling data suggest that cannabinoids suppress T-cell function and that CBR-1 and CBR-2 signaling also plays a role in the regulation and magnitude of T-cell response (Table 1).

#### 4.1.1. T-Regulatory (T-Reg) Lymphocytes

T-reg is a subpopulation of T cells that regulate and suppress immune system activity expressing CD4, CD25, and FOXp3 [123]. These cells inhibit naive T cells from proliferation and differentiation into effector T cells [123]. In response to low-level T cells activation with suboptimal PMA stimulation, CBD (10 μM) induces functional T-regs that robustly suppress responder T-cell proliferation, demonstrating that the mechanism by which CBD is immunosuppressive under low-level T-cell stimulation involves induction of functional T-regs (Table 1) [124]. THC treatment could protect mice from concanavalin A (ConA)-mediated acute autoimmune hepatitis by increasing the number of FOX P3 cells in the liver [125]. Recent studies suggested that CBR-2 selective agonist helps to prolong graft survival in transplant patients (Table 1) [126,127,128]. CBR-2 activation suppresses T-cell activation, increases T-regs by decreasing the active nuclear forms of the transcription factors NF-κB and NFAT in T cells, and increases IL-10 expression (Table 1) [126].

#### 4.1.2. Natural Killer (NK) Cells

Natural killer cells, commonly referred to as NK cells, play an essential role in coordinating inflammation in the body. NK are cytotoxic lymphocytes that provide rapid responses against virally infected cells and cancer cells [129]. NK cells have been shown to express both CBR-1 and CBR-2 and release high levels of AEA and 2-AG [66]. Mice lacking CBR-2^−/−^ exhibited increased numbers of pulmonary NK cells [130]. NK cells additionally express high levels of GPR55 when an ex vivo study of human NK cells suggested that GPR55 induces the activity of NK cells by increasing the production of IFN-γ and TNF-α and Granzyme-B and CD-107 dependent cytotoxicity [131]. Similarly, an observational study on high school and university students smoking cannabis demonstrated a significant decrease in the number of functionally different subsets of peripheral blood mononuclear lymphocytes, T and B lymphocytes, mostly NK cells among cannabis smokers (Table 1) [132]. THC suppresses cloned cell lines with NK cell activity ex vivo (Table 1) [133]. THC injection in mice leads to the splenic suppression of NK activity in vivo without diminishing NK cell viability or binding to the target cells (Table 1) [134]. THC inhibited mouse and human NK cell activity without affecting their proliferation when CBRs are involved in the complex network mediating NK cytolytic activity (Table 1) [135,136].

#### 4.1.3. Cytotoxic T Cells (CTLs)

Cytotoxic T cells (CTLs) are a critical component of the adaptive immune system, providing immune defense against intracellular bacteria, viruses, and malignant cells. The vital mechanistic machinery of CTL killing involves direct exocytosis of cytoplasmic granules containing toxic factors, such as perforin and granzymes, gaining access to the target cell by receptor-mediated endocytosis and initiating apoptosis typically by granzyme-B activity and the activation of caspases [137]. Among cancer patients, cannabis use of different phytocannabinoids is predominant. The effect of phytocannabinoids on CTLs has major clinical significance using cannabis extracts or oils.

THC was shown to suppress the cytolytic activity of CTLs, inhibit the proliferation of lymphocytes responding to allogeneic stimulation, and inhibit their maturation to CTLs independently of CBR-1_1_ and CBR-2 (Table 1) [138]. Similar to NKs, THC treatment of mature CTLs did not alter the binding capacity of these cells to the target, but significantly reduced the maximal killing capacity (Table 1) [139].

The effect of THC during viral infections in animal models also shows that THC increases viral load through the decreased recruitment of CD4(+) and CD8(+) T-cells to the lungs in a murine model of influenza [140]. In another murine model, HSV-1, THC administration suppresses CTLs’ mediated lysis of infected syngeneic cells [141]. Similarly, the administration of CBD caused a significant decrease in the total leukocyte number and a substantial reduction in absolute numbers of T, B, and both T-helper and CTL subsets [142].

These in vitro and in vivo research findings need to be carefully examined and cautiously interpreted after additional extensive studies of human cells, alongside further supporting clinical testing and experiments; it strongly implies that cannabis and its active substances suppress CTL functions. Together with their effect on NK cells, the inhibition of CTLs by cannabis and cannabinoids constitutes a considerable hazard when used or consumed by immunocompromised patients with viral infections or, particularly, malignancies.

### 4.2. B Lymphocytes

B lymphocytes produce antibodies against antigens and act as antigen-presenting cells [143]. Even though B cells express the highest levels of CBR-2 among immune cells, there are minimal data and only a handful of studies in which B cells are identified as targets of ECS [98], apart from CBD-induced apoptosis B cells (Table 1) [144]. Treatment of B cells with IL-4 and anti-CD40 elevated surface CBR-2 levels [145]. CBR-2 activation of mouse splenic B cells shifted the immunity toward Th-2 type by inducing B cells class switching from IgM to IgE [146]. Notably, CBD has a suppressive effect on antigen-specific antibody production and the functional activity of splenocytes in ovalbumin-sensitized BALB/c mice, suggesting that CBD suppresses humoral immunity by impairing splenocytes functions (Table 1) [147]. The clinical validation and examination of human cells will first be needed, testing the effect of CBD, as an important implication in light of the COVID-19 pandemic and vaccination.

### 4.3. Neutrophils

Neutrophils are the most abundant granulocytes or polymorphonuclear leukocytes (PMN, PML, or PMNL). The neutrophils make up to 55–70% white blood cells. They are the first responders to infection, and they engulf bacteria by phagocytosis and secrete a range of proteins with antimicrobial effects and ROS (reactive oxygen species). CBD is a potent inhibitor of human neutrophils’ migration [148,149]; in a recent study, GPR55 was found to be expressed in human neutrophils, thus showing that GPR55 limits the tissue inflammatory responses mediated by CBR-2, while it synergizes with CBR-2 in recruiting neutrophils to sites of inflammation [149]. CBR-2 activation protects organs from several diseases, such as ischemic brain injury, arthritis, colitis, and other inflammatory diseases, by reducing the number of migrating neutrophils to the site of infection and reducing the secretion of neutrophil antimicrobial components, such as NE, MMP-9, and MPO (Table 1) [150,151]. A recent in vivo corneal injury model showed that CBD reduced inflammation by reducing neutrophil infiltration to the corneal (Table 1) [152].

### 4.4. Monocytes and Macrophages

Monocytes and macrophages are vital components belonging to the innate immune system involved in initiating, regulating, and resolving many inflammatory disorders, while, at the same time, also participating in tissue repair, wound recovery, and regeneration processes. Monocytes are highly plastic and heterogeneous and change their functional phenotype in response to environmental stimulation. Monocytes are able to differentiate into inflammatory or anti-inflammatory subsets when exposed to tissue damage or infection; then, they rapidly reach the site of injury or infection and differentiate into tissue macrophages or dendritic cells [153]. ECS signaling regulates monocytes’ and macrophages’ responses to infection. Human alveolar macrophages removed from marijuana smokers showed a reduction in phagocytosis and cytokine production in response to LPS (Table 1) [154]. As previously mentioned, CBR-2 is expressed in high levels in human monocytes and macrophages and seems to play a significant role in their functions [58,155]. As such, CBR-2 stimulation regulates dectin-1-mediated macrophage phagocytosis [156], monocyte migration and activation [157], chemotaxis (Table 1) [158], and adhesion [159]. CBR-2 stimulation of primary human macrophages also results in the decreased expression of LPS-regulated genes [160]. Interestingly, CBR-1 and CBR-2 impose opposing stimuli on peritoneal macrophages’ ROS production when the induction of ROS is CBR-1-dependent, and the activation of CBR-2 negatively regulates the process (Table 1) [161].

Therefore, it would appear that promoting pro-inflammatory responses are mediated by CBR-1 when the activation of CBR-2 dampens them. Thus, blocking CBR-1 activation or stimulation of CBR-2 could attenuate and mitigate exaggerated macrophages’ inflammatory responses. Respectively, CBD can reduce TNF-α and IL-6 secretion by peritoneal mouse macrophages stimulated ex vivo with LPS (Table 1) [162,163], or robustly induces the apoptosis of human primary monocytes (Table 1) [164] as an example of cannabis anti-inflammatory and immunosuppressive properties.

### 4.5. Dendritic Cells (DCs)

Dendritic cells (DCs) are antigen-presenting cells linking innate and adaptive immunity. DCs recognize all surrounding antigens, then process and present them to T cells to initiate the adaptive immune responses [165]. Given that DCs express both CBR-1 and CBR-2, it is reasonable to predict that the ECS plays an essential role in regulating DCs’ biology [166]. THC induces NF-kappaB dependent apoptosis of DCs through CBR-1 and CBR-2 ex vivo [104]. THC impairs antigen uptake, ex vivo, and reduces DCs CD40 and CD86 [167] and MHC class ll expression (Table 1) [168], contributing to the reduced capacity of DCs to activate T cells in vivo. Likewise, THC treatment of legionella–pneumonia-loaded DCs leads to the poor stimulation of culture-primed splenic CD4 T cells (Table 1) [167]. As follows, THC was then shown as an inhibitor of Th1 activation by targeting essential DCs functions, such as IL-12 p40 secretion, maturation, and expression of costimulatory and polarizing molecules (Table 1) [167]. Finally, the cannabinoid analog WIN55212-2, a full CBR-1 agonist with a much higher affinity than THC, inhibits inflammatory signaling pathways and promotes autophagy and metabolic rewiring of human DCs, providing in vivo protective and anti-inflammatory effects in LPS-induced sepsis (Table 1) [169]. Collectively, related to DC, compelling evidence demonstrated the potential therapeutic effects of cannabinoids treatment for human diseases involving aberrant inflammatory processes.

### 4.6. Eosinophils and Mast Cells

Eosinophils are a minority circulating granulocyte involved in allergic reactions and parasitic infection [170]. CBR-2 agonists induce the migration and chemotaxis of eosinophils in the pulmonary inflammation mice model [171]. Furthermore, CBR-2 is suggested to contribute to the eosinophil-driven disease pathogenesis (Table 1) [171]. Therefore, the antagonism of CBR-2 is suggested as a novel pharmacological approach for treating allergic inflammation and other eosinophilic disorders [171].

Mast cells are mainly found in tissues that interface with the external environment, such as the airways and gastrointestinal tract. The cytoplasm of mast cells contains large granules that store inflammatory mediators, such as histamine, heparin, and various cytokines [172]. In vitro stimulation of mast cells with endogenous and exogenous cannabinoids decreased guinea pig mast-cells activation in a CBR-2-dependent manner (Table 1) [173]. Mast cells are also reported to express CBR-1. The stimulation of this receptor limits human mast-cell activation and maturation in the skin, which may be a strategy for treating allergic and other inflammatory diseases [174]. Another confirmation of CBR’s role in mast-cell biology comes from using synthetic agonists of CBR-1 and CBR-2, ACEA, and JWH-015; both synthetic cannabinoids reduced mast-cell numbers and activation in granulomatous tissue [175].

### 4.7. Myeloid-Derived Suppressor Cells (MDSC)

Myeloid-derived suppressor cells (MDSC) are a heterogeneous population of myeloid origin cells. These cells result from altered myelopoiesis in response to the presence of a tumor or inflammation and hold potent immunosuppressive properties [176]. They play a vital role in the tumor microenvironment favoring cancer progression involved in anti-tumor immunity evasion and can impact the metastatic process by affecting anti-cancer immunity, the epithelial–mesenchymal transition, cancer stem cell formation, angiogenesis, in establishing a premetastatic niche, and supporting cancer cell survival and growth in metastatic sites [177]. MDSC presents in healthy individuals in low numbers, expanding substantially in cancer and pathological conditions associated with chronic inflammation or stress [178]. These cells have emerged as an essential contributor to tumor progression. At the same time, accumulated evidence demonstrates that the enrichment and activation of MDSC correlate with tumor progression, relapse, and adverse clinical outcomes when frequencies of circulating MDSC are suggested as a predictive marker of treatment response, as well as hindering the anti-cancer activity of immune checkpoint inhibitors [179]. Immune suppression by MDSC is mainly antigen specific, contact dependent, and utilizes several major pathways, including the production of reactive nitrogen and oxygen species (nitric oxide (NO), ROS, and peroxynitrite (PNT)), elimination of crucial nutrition factors for T cells from the microenvironment (L-arginine, L-tryptophan, and L-cysteine), disruption of T-cell homing (through the expression of ADAM17), production of immunosuppressive cytokines (TGF-β, IL-10), and induction of T-reg cells [176,180,181]. By migrating to the tumor, MDSC elevates Arg1 and iNOS, downregulating ROS production, upregulating PD-L1 on MDSC surface, and producing CCL4 and CCL5 chemokines to attract T-regs, strongly suppressing the activity of CTLs and NK cells in the tumor microenvironment [182,183]. Therefore, monitoring and eliminating MDSCs are now clinically tested to improve advanced cancer patients’ response rates to immunotherapy and better survival [184,185,186].

Overall, it seems that phytocannabinoids potently affect MDSC biology. For example, mice administrated with THC massively and rapidly expanded MDSC, expressing functional arginase and exhibiting potent immunosuppressive properties in vitro and in vivo, associated with an increase in granulocyte colony-stimulating factor (G-CSF). Using CBR-deficient mice or antagonists to CBR-1 and CBR-2 demonstrated that these processes are indeed mediated by both CBR-1 and CBR-2 (Table 1) [187]. Furthermore, THC-mediated epigenetic modifications elicited MDSC activation by increasing the promoter methylation of DNMT3a and DNMT3b, reducing their expression. Likewise, Arg1 and STAT3 promoters’ methylation was decreased in THC-induced MDSC, resulting in higher levels of Arg1 and STAT3 expression (Table 1) [188]. However, most significantly, THC enhances the growth and metastasis through the induction of MDSC in murine breast cancer [189]. Additional support that CBRs and cannabinoids play a role in MDSC biology came from the observations that once administered to naive mice, a peritoneal injection of CBD triggered a robust induction of MDSC (Table 1) [190]. CBD-induced MDSC comprised of granulocytic and monocytic MDSC subtypes, monocytic MDSC exhibited strong T-cell-suppressive function. Again, using CBR antagonism or animals with CBR deficiency demonstrated their crucial role in immune regulation via the induction of MDSC [190]. Along the same line, the in vivo experimental administration of CBD decreased autoimmune hepatitis by inducing the accumulation of MDSCs in the liver when depleting MDSCs with anti-Gr1 mAb neutralized the protective effect of CBD. Using Trpv1-deficient mice, it was shown that CBD acts independently of TRPV1 [191], although, in contrast, in a similar study, CBD triggered MDSC, attenuating acute inflammation in the liver via the TRPV1 receptor [192].

Using animal models and not directly from tumors, MDSC monitored in the circulation or peritoneum still strongly suggested that cannabis consumption, particularly the cannabinoids THC and CBD, may affect tumor progression negatively and interfere with anti-tumor immunity by inducing and activating suppressive MDSCs. Significant links to MDSC biology should be addressed and it is paramount to clinically characterize cannabis’s impact on the immune cell population to substantiate clinical implications. We believe that the clinical characterization of cannabis’s effect on effector immune cells may prevent adverse and non-desirable clinical outcomes.

## 5. Conclusions and Perspectives

Over the years, there has been a broad consensus on the extensive and potent immunosuppressive properties of two main cannabis cannabinoids (i.e., THC and CBD) and cannabis-based products (Figure 1 and Figure 2). Basic ex vivo studies using isolated cells, animal model research studies, and clinical data demonstrate that cannabis mediates its immunosuppressive influence on the immune system effector cells using three main mechanisms: (i) it imposes apoptotic cell death in a wide range of immune cells [124,168,171]; (ii) it suppresses pro-inflammatory cytokine production and antimicrobial components by activated cells [121,122,154,155]; and (iii) it inhibits immune effector cell activation and functions [149].

**Figure 2 pharmaceutics-14-00389-f002:**
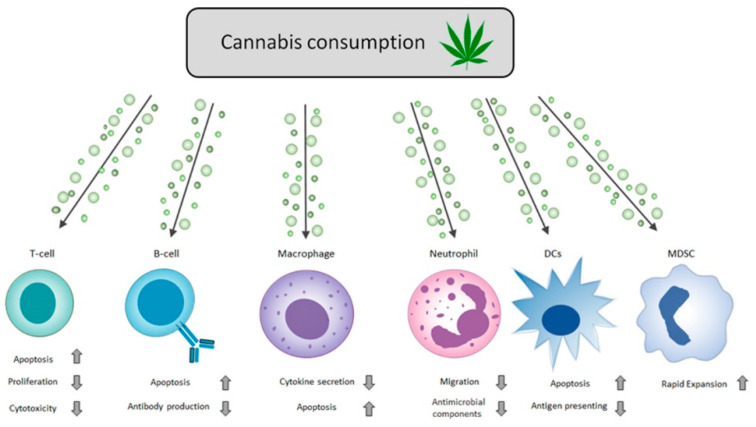
Cannabinoids regulate immune responses. Schematic diagram of cannabis immuno-modulation properties on different effector cells of the immune system. The arrows in the chart above represent the adjacent cellular process’s increase (**up arrow**) and decrease (**down arrow**). Cannabis consumption expressed above includes all forms of cannabis-based products (smoking, ingestion, or the direct use of THC and CBD). For specific details on each component and its effects, refer to the review’s specific immune cell type section. See Table 1 for a detailed summary of the effects of particular cannabinoids on specific immune cells.

**Table 1 pharmaceutics-14-00389-t001:** The effects of cannabinoids and agonists/antagonists of CBR-1/2 on immune cells.

Component	Cell Type/Animal Model	Experimental Design	Effects	Reference
THC	Human T cells, B cells, and DCs	Ex vivo	Increased apoptosis	[102]
THC	BALB/c mice	Ex vivo	Suppressed the proliferation of spleen and lymph node cells	[116]
THC/CBD	Human T cells, B cells, eosinophils, CD-8, and NK cells	In vitro	Decreased cytokine production	[106]
THC	Splenocytes derived from BALB/c mice	Ex vivo	Decreased the production of Th1-associated cytokines, including g IL-2, IL-12, and interferon-g (IFN-g), increasing the production of Th2 related cytokines, such as IL-4 and IL-10	[117]
THC	Splenocytes from C3H/HeJ mice	Ex vivo	Suppressed cloned cell line with NK-cell activity	[133]
THC	Splenocytes from C3H/HeJ mice	In vivo and ex vivo	Suppressed NK-cell activity	[134]
THC	Splenocytes from C57Bl/6 mice or CB1^−/−^ CB2^−/−^ C57Bl/6	Ex vivo	Suppressed the function of CTLs independent of CB1-R and CB2-R	[138]
THC	C3H/HeN mice or splenocytes derived from C3H/HeN mice	In vivo and in vitro	Suppressed the induction and cytolytic activity of CTLs	[139]
THC	Murine peritoneal macrophages (B6C3)F1 and C57BL/6 mice	Ex vivo	Inhibited migration (CB-R dependent)	[158]
THC	Bone marrow-derived cells from C57BL/6 mice	Ex vivo	Induced the NF-kappaB-dependent apoptosis of DC’s through CBR-1 and CBR-2	[104]
THC	Bone-marrow-derived cells from BALB/c mice	Ex vivo	Inhibited the expression of MHCII and costimulatory molecules CD40 and CD86 and poor stimulation of CD4 T cells to legionella–pneumonia-loaded DCs	[167]
THC	MDSC cells derived from BL6 mice	In vivo	Induced MDSCs by epigenetic changes that reduce the expression of DNMT3a and DNMT3b and increased the expression of Arg1 and STAT3	[188]
THC	C57BL/6 mice or purified peritoneal CD11b+Gr-1+ cells from C57BL/6	In vivo and in vitro	Induced MDSC cells and their expansion to the periphery by increasing G-CSF dependently on CB1-R and CB2-R	[187]
CBN	EL-4 cell line	In vitro	Increased IL-2 in activated T cells (CB1-R and CB2-R independent)	[112]
CBD	Splenocytes derived from C57BL/6 mice	In vitro	Induced CD4+ CD25+ T-regs to robustly suppress responder T-cell proliferation	[124]
CBD	Male BALB/c mice	Ex vivo	Increased apoptosis of thymocytes by increasing ROS generation	[120]
CBD	EL-4 thymoma cell line	In vitro	Increased apoptosis by increasing ROS generation	[120]
CBD	Male adult Wistar rat	In vivo	Reduce production of serum cytokines	[105]
*Cannabis sativa* L. extract 5% CBD	Human neutrophil	In vitro	Decreased migration, ROS generation, and TNF-α production	[107]
CBD	Splenic T cells derived from B6C3F1 or C57BL/6 mice	Ex vivo	Reduced IL-2 and IFN-γ cytokines production	[112]
CBD	Splenocytes derived from C57BL/6	Ex vivo	Increased apoptosis of T cells and B cells	[144]
CBD	BALB/c mice or splenocytes derived from BALB mice	In vivo and ex vivo	Suppressed antigen-specific antibody in OVA-sensitized mice and decreased production of IL-2, IL-4, and IFN-γ	[147]
CBD	Peritoneal macrophages derived from NOD/LtJ mice (model for diabetes)	Ex vivo	Reduction in plasma levels of the pro-inflammatory cytokines, IFN-g and TNF-a and increased levels of anti-inflammatory cytokine IL-4 and IL-10.	[162]
CBD	Human monocytes	Ex vivo	Increased apoptosis	[164]
CBD	Female C57BL/6 mice and C3H/HeJ mice	In vitro	Induction of immunosuppressive CD11b+ Gr-1+ MDSC in naive mice dependently on mast cells and primarily mediated by PPAR-g	[190]
Delta8-THC, CBD, and CB2-R agonist (HU-308).	BALB/c and CBR2^−/−^ mice (model for corneal injury)	In vivo	Reduced neutrophil infiltration to the cornea. The anti-inflammatory effect of delta8-THC is dependent on CBR-1, whereas that of CBD and HU-308 is dependent on CB2-R.	[152]
CB2-R agonist (JWH-015 and JWH-133)	Human monocytes	Ex vivo	Reduced chemotactic response	[157]
CB2-R agonist (O-1966)	Splenocytes from WT C57BL/6 mice or CB2-R knockout (CB2-R k/o)	Ex vivo	Decreased levels of NF-κB and NFAT, and increased levels of IL-10 expression in WT T cells, but not in T cells from CB2R k/o mice. Additionally, increased levels of T-regs.	[126]
CB1-R antagonist (SR 141716) /CB2-R (SR144528) antagonist	Splenocytes derived from Male Swiss mice	Ex vivo	Reversed the INF-(g) reduction in NK cells induced by delta9-THC	[136]
CB2-R agonist (JWH-133)	Bone-marrow-derived cells from male C57BL/6 mice and CB1^−/−^ and CB2^−/−^ mice	Ex vivo	Inhibited neutrophil recruitment to the brain and protection against ischemic brain injury	[150]
CB1-R agonist (ACEA), CB2 (JWH015), and antagonists of both receptors	Peritoneal macrophages derived from C57BL/6J WT and CB2^−/−^ mice or RAW264.7	In vitro and ex vivo	CB1-R agonist increased ROS production and activation of CB2-R negatively regulated the process	[161]
CB1-R agonist (ACPA) and antagonist (AM251)	Bone-marrow-derived cells from C57BL/6 mice	Ex vivo	ACPA showed a reduction in MHC-II cell surface expression and reduced the T cell stimulatory capacity of DC	[168]
CB1-R agonist (WIN55212-2)	Human DCs	Ex vivo	Inhibits inflammatory signaling pathway and promote autophagy	[169]
CB2-R agonist (JW-113)	C57Bl6/N mice (model of lung inflammation)	In vivo	Induced migration and chemotaxis of eosinophils	[171]
CB agonist CP55,940	Mast cells derived from Dunkin-Hartley guinea pigs	Ex vivo	Decreased mast cell activation in a manner dependent on CB2-R receptor	[173]
Bhang (marijuana)	Human NK cells, B cells, and, T cells derived from smokers as compared to control	In vivo	Significant decrease in the number of functional cells	[132]
Marijuana	Human alveolar macrophage cells derived from smokers	Ex vivo	Impaired alveolar macrophage function and cytokine production (g TNF-α and TGF-β)	[154]

Well-documented studies over the years and constantly emerging new data have made medical cannabis a popular palliative tool for a spectrum of pain relief treatments with its therapeutic medical properties [16]. Cannabis is widely used in oncology as an effective tool to ease the treatment and chemotherapy process and other cancer-related side effects [90]. Although cannabis has many benefits, “one side of the coin”, it may also cause considerable health hazards with detrimental effects, “another side of the coin”. For example, the integration of cannabis with anti-cancer immunotherapy treatment reduces the response rate to nivolumab and correlates with higher mortality rates among cannabis users than in non-users when immunotherapy is applied [108,109]. Several reports suggest that exposure to cannabinoids can increase bacterial and viral infection susceptibility due to reduced inflammatory effects. While less common, cannabis or cannabinoids have been shown to increase infection mortality. The daily administration of THC to mice led to increased susceptibility to Legionella pneumophila infection [193], decreased T-cell proliferation, increased lesion severity, mortality by herpes simplex virus type-2 (HSV-2) infection [194], increased influenza virus load, diminished immune cell recruitment to the lungs [195], activation of the lytic switch of Kaposi’s sarcoma-associated herpesvirus (KSHV) [196], increased viral titers of vesicular stomatitis virus (VSV) [197], weak antibody production with no neutralizing antibody generation against cowpox [197], and rapid progression of fibrosis due to chronic hepatitis C (HCV) [198]. However, in another study, cannabis use was associated with a decreased incidence of liver cirrhosis among HCV patients [199]. THC can also suppress immune function, increase HIV coreceptor expression, and cofactor to enhance human immunodeficiency virus (HIV) replication [161].

Immensely controversial with conflicting results by clinical experiments, cannabinoids’ impact on the immune system is beneficial in dampening chronic or unrestrained persistent inflammatory conditions, such as autoimmune diseases. CBD effectively reduces inflammation in rheumatoid arthritis (RA) patients and possesses anti-arthritic activity, which may alleviate arthritis via targeting synovial fibroblasts under inflammatory conditions [200,201,202]. Cannabis’s anti-inflammatory effects are beneficial in treating inflammatory bowel diseases, such as colitis, experimentally induced colitis resulting in reduced colon injury, iNOS, and IL-1β expression, and all by CBD treatment [203]. We propose that patients with ongoing infections refrain from regular cannabis use.

Looking at both sides, cannabinoids and their positive and effective influence on patients’ quality of life are mechanisms, “one side of the coin”, while the same cannabinoids treatments affect the immune system, “the other side of the coin”. Undeniably, cannabis compounds need extensive research and clinical validation for researchers to attain a comprehensive understanding of the circumferential effects and the molecular, biochemical, and primarily biological mechanisms by which cannabis moderates the immune system. While we firmly believe in cannabis as an efficient palliative tool and in the potential of cannabinoids-based drugs or the ECS components as drug targets to be fully integrated into clinical practice, we urge the medical community to acknowledge its hazardous effects and immunomodulation properties to ensure the minimization of unwanted adverse effects when combining cannabis alongside medical treatments.

## Figures and Tables

**Figure 1 pharmaceutics-14-00389-f001:**
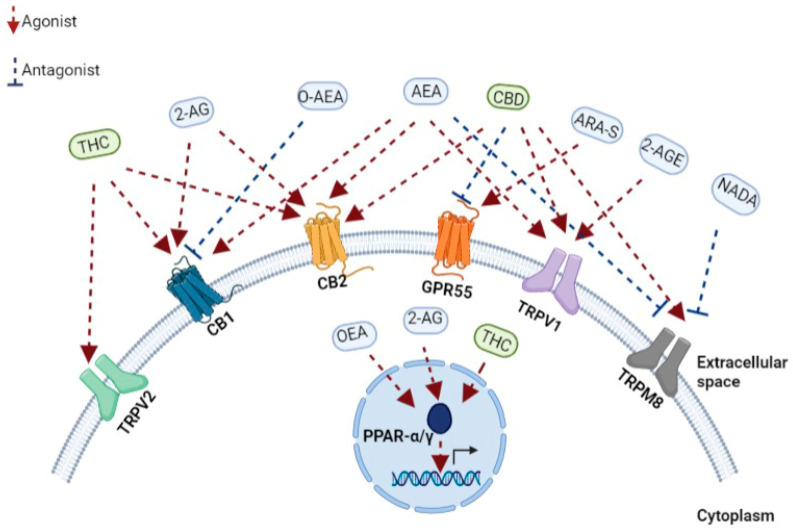
A schematic representation of the main components of the endocannabinoid system. Receptors, endogenous cannabinoids, and cannabis-based components. THC, tetrahydrocannabinol; CBD, cannabidiol; 2-AG, 2-arachidonoylglycerol; O-AEA, virodhamine; AEA, anandamide; ARA-S, N-arachidonoyl serine; 2-AGE, 2-arachidonoyl glycerol ether; NADA, N-arachidonoyl dopamine; TRPV2, transient receptor potential cation channel subfamily V member 2; CB1, cannabinoid receptor 1; CB2, cannabinoid receptor 2; GPR55, G protein-coupled receptor 55; TRPV1, transient receptor potential cation channel subfamily V member 1; TRPM8, transient receptor potential cation channel melastatin 8; PPAR-α, peroxisome proliferator-activated receptor-alpha; PPAR-γ, peroxisome proliferator-activated receptor-gamma.

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
