# Peer review of "“The Two Sides of the Same Coin”—Medical Cannabis, Cannabinoids and Immunity: Pros and Cons Explained"

_pharmaceutics, 2022, doi:10.3390/pharmaceutics14020389_

Round 1

Reviewer 1 Report

The manuscript compiles the literature on the medicinal properties of Cannabis into a useful review. The intended focus of the review is the contrast (or the two-sides) of the established benefits against suppression of the immune system.  Evidence that cannabinoids suppress the function of different immune cell types are presented in individual sections and simplistically represented with the only graphic in the manuscript. The epidemiological studies on the detrimental effects of cannabis use in various illnesses is presented in the conclusion. Having all of this information compiled into a review is very useful, but it could be presented more effectively with the following suggestions:

  1. It would be more effective to present the epidemiological information early in the manuscript, before going into the individual cell types, because without this information the question of whether the presented studies are relevant to the real world is not answered until the very end.
  2. In vitro apoptosis studies are questionable because one has to consider if the applied concentration of cannabinoids and other cell culture conditions are a realistic reflection of what happens in vivo. However the graphic represents apoptosis as an equally significant finding to the more conclusive animal model and isolated cell-based assay experiments. A more descriptive table or graphic can convey the information more accurately.
  3. Additional graphics would help clarify the information. One important addition would be a representation of the Endocannabinoid system including the various endogenous and marijuana derived ligands would be a great way to summarize some of the information in the introduction.
  4. Organization of the individual sections could be better, a lot of information is presented in each paragraph but there is no cohesive narrative.

Author Response

We sincerely thank you for all your hard work, professionalism, and time dedicated to reviewing our manuscript. Your insights and suggestions were very valuable.

Your suggestions and input from a fresh outside perspective have made the manuscript much better.

Specifically, we adopted Reviewer 1 recommendations and edited Table -1 to include additional graphics to help clarify the information and represent the Endocannabinoid system, including the various endogenous and marijuana. Also, as proposed, we added highlighted and stressed, where relevant, information regarding in-vivo or in-vitro methodology. In particular, we broadened the informational text regarding apoptosis, explaining in-vivo or in-vitro in each experiment. Nonetheless, we managed to moderate the statements as suggested. Finally, as indicated by Reviewer-1, we added Figure-1 to clarify the information and representation of the Endocannabinoid system and its various endogenous and marijuana-derived ligands.

As for Reviewer-2, we profoundly edited and modified the manuscript according to his remarks and suggestions. We added many needed citations, text rephrasing, clarifications, and comprehensive  English language and style editing. Regarding Figure-1, now Figure-2, we added a reference to Table-1 for specific details regarding each process. A better clarification explaining the 'two sides' is now added, and we believe it is now more comprehensive.  

Reviewer 2 Report

This papers thoroughly reviews the immunomodulatory effects of cannabis and cannabinoids, as well as the mechanisms involved. It highlights potential related to such a use when immunity is compromised. A great amount of work has evidently been involved in this manuscript.

As general comments I would suggest :

  • a thoroughly re-reading by a native English-speaker person.
  • checking for irrelevant use of upper cases
  • to clearly present all results that illustrate (or contradict) authors' point coming from studies on HUMAN (clinical or observational studies). I suggest a dedicated table. Indeed, data on human are diluted into different parts dedicated to molecular/cells mechanisms. It seems important for me to clearly adress and present what is currently available in Human to assess the need for further research and the extent to which such putative mechanims translate into real harms.
  • A clarification of the 'two sides' would be beneficial. It is not clear if it is 'beneficial' vs. 'harmful' cannabis/oids (generally speaking) or 'beneficial' vs. 'harmful' immunomodulatory properties of cannabis/oids.
  • Figure 1 is great. Maybe (just a suggestion) could you add on the long arrows coming from cannabis leaf the mechanims invloved (by indicating the molecules (THC, CBD) and/or receptors involved)
  • Please find more comments within the PDF attached.

Author Response

(The authors gave the same response as above.)

Round 2

Reviewer 2 Report

I thank the authors for the revisions they made. The manuscript has gained in quality.

I only have few comments attached in the PDF.

Author Response

We highly appreciate the substantial reports of Reviewer-2 at the first round, and it is additional time on the second round looking after our manuscript.  This would help us promote this critical message, which clinicians and researchers frequently overlook.

All three additional remarks were adopted and are now incorporated into the text.

  1. "Cannabis can also be combusted and inhaled alone without tobacco" is now being added where indicated.
  2. Cannabis combustion involves exposure to harmful substances-

the following sentence is now incorporated where indicated- "However, cannabis combustion is considered more problematic and less favorable than vaporization due to exposure to harmful toxic substances created by combustion."

  1. Table-1 was restructured as suggested- First by component and the by cell type in each section. A new Table-1 is attached
  2. Finally, the manuscript was sent for professional English editing- The minor text changes of the editor appear as word track changes.  

Including all changes and suggestions, we genuinely believe the text is now ready for publication